# Charitable crowdfunding donation-intention estimation depending on emotional project images using fNIRS-based functional connectivity

**SuJin Bak**[1⊛], **Minsun Yeu**[2⊛], **Dongwon Min**[3]*, **Jaehoon Lee**[4]*, **Jichai Jeong**[5]*

**1** Advanced Institute of Convergence Technology, Suwon-si, Gyeonggi-do, Republic of Korea, **2** College of Business Administration, University of Ulsan, Ulsan, Republic of Korea, **3** College of Business, Dankook University, Yongin, Gyeonggi, Republic of Korea, **4** Department of Computer Science and Engineering, Korea University, Seoul, Republic of Korea, **5** Department of Brain and Cognitive Engineering, Korea University, Seoul, Republic of Korea

⊛ These authors contributed equally to this work.
* dwmin@dankook.ac.kr (DM); ejhoon@korea.ac.kr (JL); jcj@korea.ac.kr (JJ)

**Data Availability Statement:** All relevant data are within the manuscript and its Supporting Information files.

## Abstract

Charitable fundraising increasingly relies on online crowdfunding platforms. Project images of charitable crowdfunding use emotional appeals to promote helping behavior. Negative emotions are commonly used to motivate helping behavior because the image of a happy child may not motivate donors to donate as willingly. However, some research has found that happy images can be more beneficial. These contradictory results suggest that the emotional valence of project imagery and how fundraisers frame project images effectively remain debatable. Thus, we compared and analyzed brain activation differences in the prefrontal cortex governing human emotions depending on donation decisions using functional near-infrared spectroscopy, a neuroimaging device. We advance existing theory on charitable behavior by demonstrating that little correlation exists in donation intentions and brain activity between negative and positive project images, which is consistent with survey results on donation intentions by victim image. We also discovered quantitative brain hemodynamic signal variations between donors and nondonors, which can predict and detect donor mental brain functioning using functional connectivity, that is, the statistical dependence between the time series of electrophysiological activity and oxygenated hemodynamic levels in the prefrontal cortex. These findings are critical in developing future marketing strategies for online charitable crowdfunding platforms, especially project images.

## Introduction

Charity fundraising increasingly relies on online crowdfunding platforms [1]. Donation-based crowdfunding platforms are easy to access and operate, allowing fundraisers to launch charity crowdfunding projects easily. Fundraisers post their projects with images and brief

**Funding:** This work was supported in part by the Ministry of Education of the Republic of Korea and the National Research Foundation of Korea (NRF-2023S1A5A8076043) and in part by the Ministry of Education of the Republic of Korea and the National Research Foundation of Korea (NRF-2021S1A5A2A01067613).

**Competing interests:** The authors have declared that no competing interests exist.

descriptions on an online charitable crowdfunding platform. As competition for charitable crowdfunding projects intensifies, research on the effect of the project image on donor behavior is needed to increase the project success effectively.

Project images use a variety of emotional appeals to promote helping behavior. According to the literature, charitable donors are influenced by project images, and emotionally engaging images are critical in attracting donor attention and motivating charitable donations [2]. Thus, when seeking an answer to what makes people perform helping behavior, it is crucial to consider the behavior concerning the emotions the project image generates.

Negative emotions are commonly used to motivate positive behavior, such as donating. Consumers often rely on victim images (i.e., beneficiary images with negative expressions) to decide to engage in helping behavior [3]. The image of a happy child may not motivate donors to donate as willingly. Most past studies have suggested that negative emotions, such as sadness, expressed in images more effectively elicit helping behaviors [4–6]. For example, Burt et al. found that images of children that evoked negative emotions produced higher positive responses to donations and potential donations than images that evoked positive emotions [5]. Fisher et al. found that, unlike negative emotions, positive emotions, such as love and pride, did not increase donation behavior in response to television advertisements focused on children's needs and fundraising [7]. Most donors donate money when the image is negative rather than happy or neutral. Negative emotions are more effective than positive emotions across texts and content, including images [8].

Moreover, the group Save the Children completed an in-house two-year research project and decided to change how they portray beneficiaries by adding positive imagery [9]. Their findings revealed that a happy victim image may be more beneficial [10]. Seeing a happy child in a charity advertisement can drive a larger potential donation because it allows donors to see the results of their donation. Sciulli et al. argued that advertisements with high positive emotional intensity encourage viewers to participate in relevant social causes [11].

These contradictory results suggest that the emotional valence of project images is still debatable, suggesting that how fundraisers frame project images effectively remains unclear. We solve this problem through empirical biosignal analysis using functional near-infrared spectroscopy (fNIRS), the latest neuroimaging device. Many studies on emotional valence can be interpreted according to the prefrontal cortex (PFC) activity of the brain. The PFC is situated in the frontal lobe toward the front of the brain. Its primary role is to govern human emotions, such as cognitive control, encompassing attention, behavior, thoughts, and so on [12]. In line with this role, studying the brain activity of the PFC enables us to anticipate human behavioral donation patterns. One approach to understanding the state of the PFC is to calculate and analyze the functional connectivity (FC). Some studies have found a tendency to predict emotional states well by analyzing the FC [13,14]. A recent study strongly supported that the strength of FC can be a useful and powerful way to explain the neuropathies of affective disorder [15].

Therefore, this study explores the emotional valence of project images related to the intention toward charity crowdfunding projects. We compared and analyzed differences in brain activities, such as FC, depending on charitable donation decisions by analyzing fNIRS signals measured while watching three emotional valence (i.e., positive vs. negative vs. neutral) project images.

## Materials and methods

### Participant demographics

Before conducting the study, we calculated the desired sample size using G*Power 3.1.9.2. Referring to Ref.[16], per the task group, the target sample size of 64 individuals was

determined based on assumptions of a moderate effect size ($\alpha = 0.05$, power = 0.8, effect size = 0.25), with all variables standardized. Thus, 64 people (32 males and 32 females) from Korea University between 20 and 33 (mean and standard deviation of age: M = 25.67 sd = 2.69 years) participated in a same-day fNIRS study. All participants were recruited from the online community around KOPAS affiliated with Korea University over the span of a month. We have explained the whole process of experimenting, and all participants fully understood it. We recorded the demographics of the participants in Table 1. Participants with sensory impairments, epilepsy, and brain injury were excluded. All subjects were right-handed and used their dominant hand in the experiment to ensure consistency in brain activation patterns. They had normal or corrected-to-normal vision. The participants had no previous history of any physical, mental, or psychological disorders. All participants provided written informed consent to participate in the study. All experimental procedures were approved by the Korea University Institutional Review Board (KUIRB-2022-0318-01). Experiments were carried out in accordance with the Helsinki Declaration's requirements. All participants were financially compensated for their time.

## fNIRS device specification and data preprocessing

A high-density fNIRS device (NIRSIT Lite; OBELAB, Seoul, Korea) measured the relative changes in oxy-Hb and deoxy-Hb. Fig 1 presents the composition of the fNIRS system, which has five dual-wavelength (780/850 nm) laser diodes and seven photodetectors, resulting in 15 channels, with each a source and detector around 3 cm apart. The optical signal variation of each channel was sampled at 8.138 Hz. We quantified the frequency band of the detected channels using bandpass filtering from 0.005 to 0.1 Hz to remove the slow drift of physiological and environmental noise. The threshold of the signal-to-noise ratio was 30 dB. Relative hemodynamic changes in each task were extracted using the modified Beer–Lambert law [17]. The baseline relative change was defined as the average value from −5 to 0 s before the start of each task period. The multitrial results were individually block averaged, and grand averaging was applied to extract representative mapping results from each group.

## Emotional preferences depending on the victim images

We instructed all participants in an online questionnaire to indicate their emotional preferences regarding the project images using a survey software program (Qualtrics, Inc., Provo, UT). All participants' feelings about the victim images were scored on a 7-point Likert scale,

**Table 1. Summary of demographic subject information.**

| Variable | Category | Number of subjects | Percentage |
|---|---|---|---|
| Race | Asian | 64 | 100.00% |
| Gender | Men | 32 | 50.00% |
| | Women | 32 | 50.00% |
| Age range | 20s | 57 | 89.06% |
| | 30s | 7 | 10.94% |
| Employment | Unemployed (including students) | 60 | 93.75% |
| | Employed | 4 | 6.25% |
| Education | High school graduate | 44 | 68.75% |
| | Bachelor's degree | 18 | 28.13% |
| | Graduate degree | 2 | 3.13% |

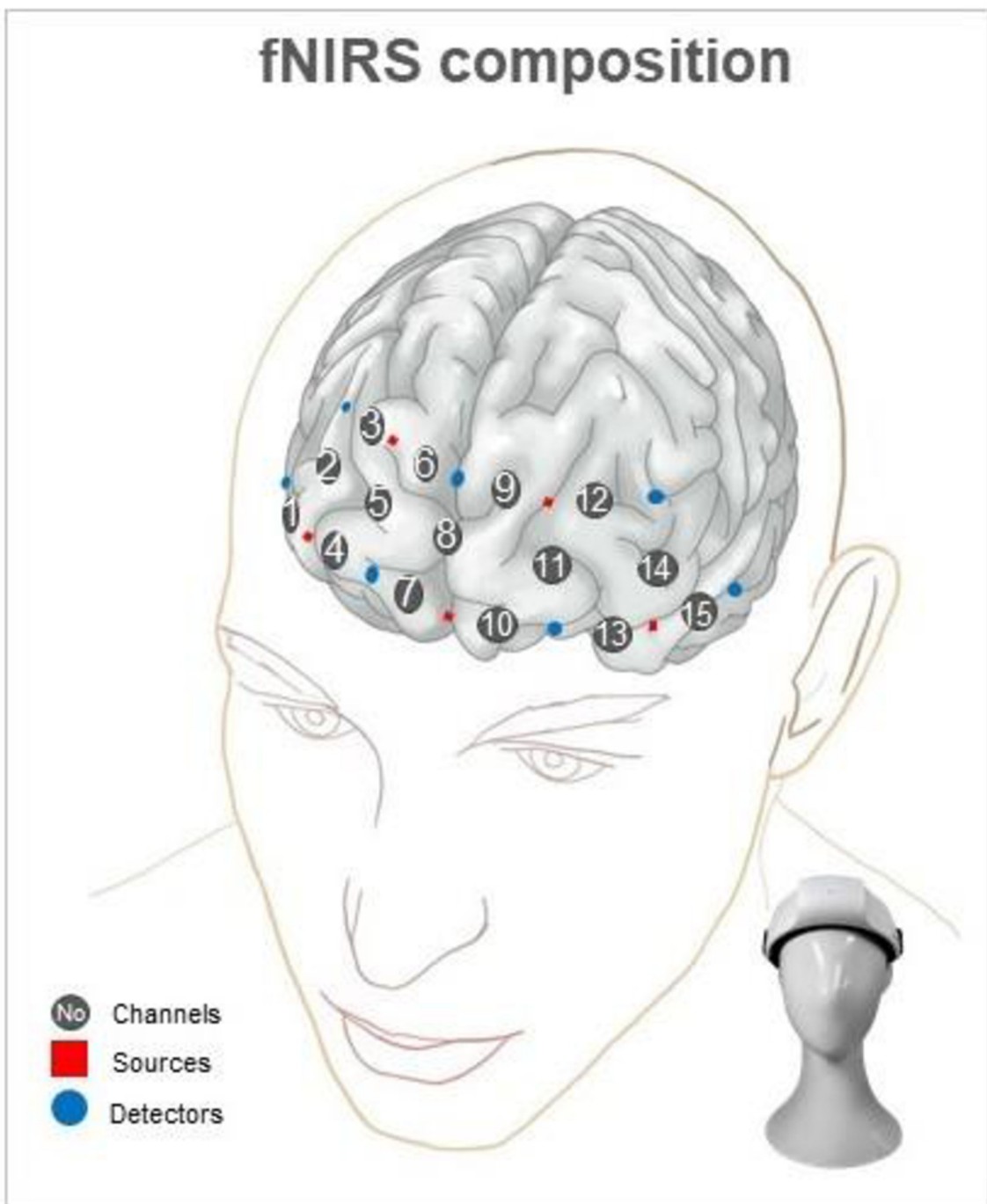

**Fig 1. fNIRS system composition in prefrontal cortex(PFC) brain areas.** Gray-filled circles indicate fifteen channels consisting of five sources (red squares) and seven detectors (blue-filled circles). Centering on Ch 8, Chs. 1–7 correspond to the right frontal lobes, and the rest correspond to the left frontal lobes.

ranging from "very negative" to "very positive," to quantify the three valence ratings (i.e., positive, negative, and neutral emotions).

### Questionnaire on whether to donate or shop

After the experiments, all subjects examined the emotional project image and determined whether to donate or shop. Their responses were graded on a 7-point Likert scale, ranging from "strongly agree" to "strongly disagree" regarding the donation or shopping activity.

### Online donation task protocol

Per the presurvey donation intentions, we divided the experimental participants into three groups: donation, no donation (shopping), and nonintention groups. Subsequently, all subjects were reportedly exposed to all three types of images (positive, negative, and neutral), as specified in the within-subject design (i.e., repeated measures design) process. This design can minimize the variance caused by individual differences among participants. Each group randomly performed the three emotional project-image-watching tasks. All project images used as donation images on Happybean were provided by the Korean internet platform Naver, a platform for nonprofit organizations to promote their projects and raise funds. We presented positive, negative, and neutral emotional images to induce incidental emotional states in participants before they made two types of donation decisions. Fig 2 represents the experimental protocol. The overall experiment consists of a cue (1 s), an image-watching task time (25 s), and a break time (30 s). Using the fNIRS device, we recorded hemodynamic responses during the online donation task protocol. Then, we divided the recorded hemodynamic responses into nine categories: Types 1 to 9. Tasks 1, 2, and 3 involve watching positive, negative, and neutral project images, respectively.

## Experimental results

### Valence rating results

We investigated the valence ratings of project images in this experiment. Table 2 lists the reactions to the project images. Of the 64 subjects, 25 people expressed that the project images in Task 1 were positive (39.06%), 33 indicated that project images in Task 2 were negative (51.56%), and 23 expressed that project images in Task 3 were neutral (35.94%). These ratios account for the largest percentage of responses for the three categories. In Tasks 1, and 2, the subjects answered the most positive, negative, and neutral images, respectively. On the other hand, Task 3 elicited predominanatly neutral responses, yet there was a notable majority

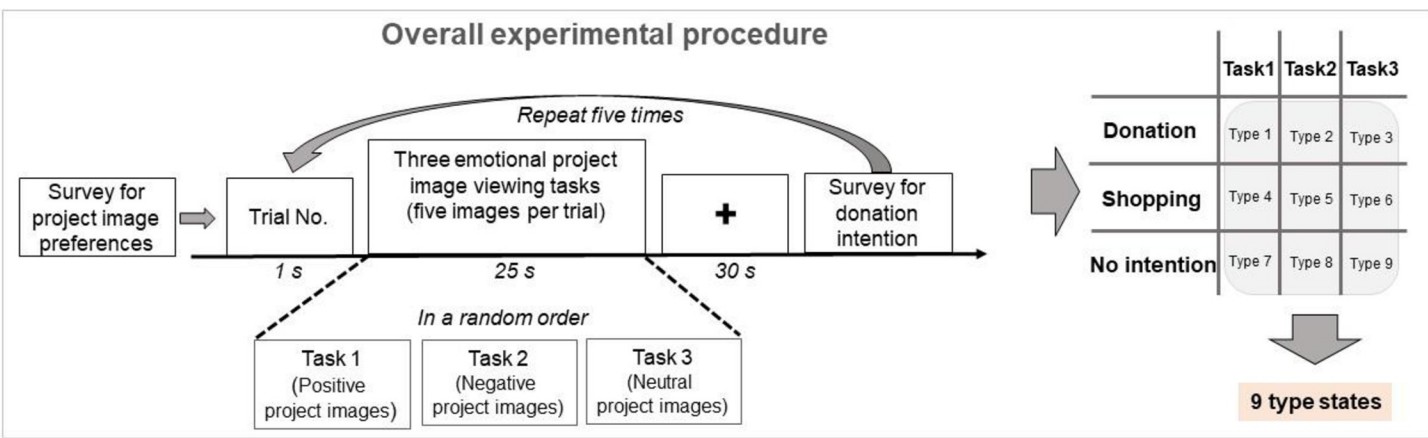

**Fig 2. Overall experimental procedure.** Tasks 1, 2, and 3 present images corresponding to positive, negative, and neutral perspectives, respectively.

**Table 2. Survey results quantifying the valence rating for emotional images in each task.**

| | Task 1 (Positive images) No. (%) of subjects | Task 2 (Negative images) No. (%) of subjects | Task 3 (Neutral images) No. (%) of subjects |
|---|---|---|---|
| Strongly positive | 6 (9.38%) | - | 5 (7.81%) |
| Positive | **25 (39.06%)** | - | 18 (28.13%) |
| Slightly positive | 13 (20.31%) | 1 (1.56%) | 17 (26.56%) |
| Neutral | 11 (17.19%) | 3 (4.69%) | **23 (35.94%)** |
| Slightly negative | 6 (9.38%) | 12 (18.75%) | - |
| Negative | 2 (3.13%) | **33 (51.56%)** | 1 (1.56%) |
| Strongly negative | 1 (1.56%) | 15 (23.44%) | - |

expressing positive feedback. In this study, we are focused on the effect of positive and negative images on donations, and the valence quantitative ratings of positive and negative tasks are considered appropriate. Thus, we conclude that the valence rating of the victim images in this study is properly designed.

## Responses on deciding whether to donate or shop

Fig 3. illustrates the survey results for the donation intentions for the three tasks. The left panel is a graph on the decision of whether to donate. Conversely, the right panel graphs the decision of whether to shop instead of donate. Unlike Task 3, with neutral images, Tasks 1 and 2 (with positive and negative images, respectively) greatly contributed to increasing donating. Task 3 encouraged people to shop instead of donating.

## PFC activity differences

We recorded the functional hemodynamic signals depending on donation intentions, as explained in Fig 4. The colored bar marks the quantifiable brain activation, ranging from -1 (low activation) to 1(high activation). The decision to donate is unaffected by emotional tasks, regardless of whether the visuals are positive or negative. No visually significant difference occurs in the presented brain activity (Fig 4). This phenomenon is similar to the donation

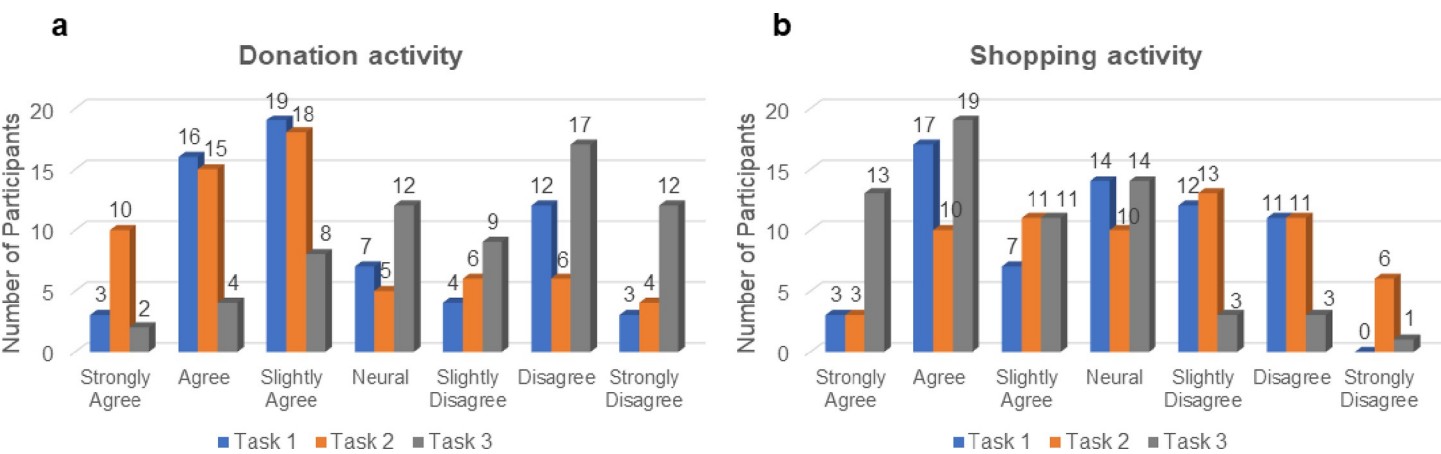

**Fig 3. Valence rating survey results.** (a) Graph depicting a donor's decision to donate or not. (b) Graph on purchasing rather than contributing.

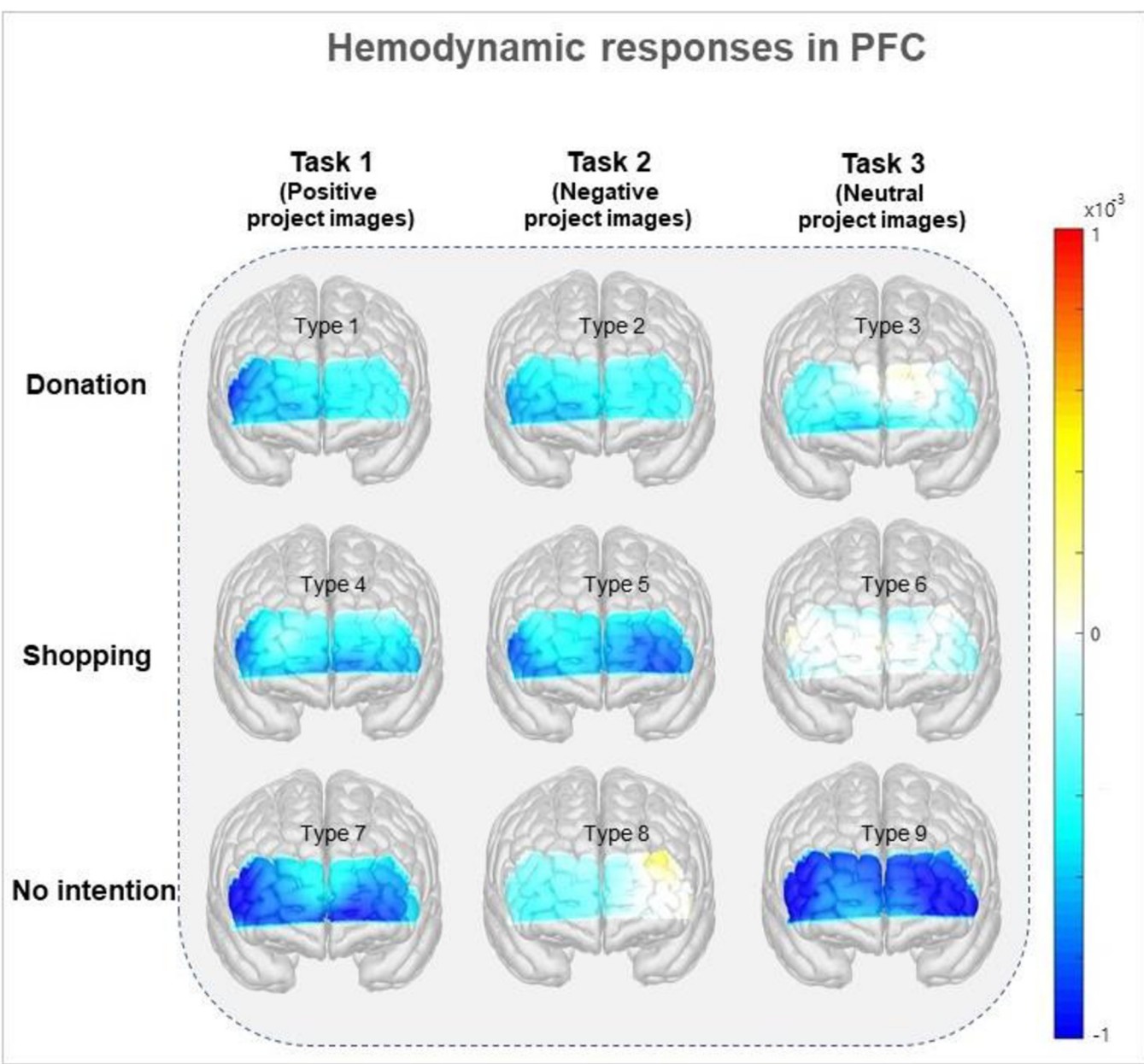

**Fig 4. Hemodynamic responses by prefrontal cortex activities.** The horizontal axis represents intentions to donate, and the vertical axis is the type of experimental task. The color bar is quantifiable brain activation ranging from -1 (low activation) to 1 (high activation). The decision to donate or not is unaffected by emotional tasks, regardless of the positive or negative visuals, suggesting that negative commercials do not necessarily result in donations.

intention survey results for the three tasks. Hence, negative victim images do not always lead to donations.

## Channelwise functional connectivity

Channelwise FC represents the statistical relationship between the 15 hemodynamic sensor signals measured from the subject's donation intention within each task activity. FC works based on statistics, which maps the correlation between responses shown in the PFC of a subject brain. The functional connectivity is a potential biomarker, estimated as the Pearson

correlation coefficient between the mean time series from each channel of interest. Pearson's correlation coefficient was used to calculate the strength of the temporal correlation of the hemodynamics between all channel combinations.

Fig 5 presents the FC results regarding the subjects' nine donation intention types. The horizontal axis represents their intention to donate, whereas the vertical axis represents stimuli-response tasks with positive, negative, and neutral images. Regardless of the task type, the

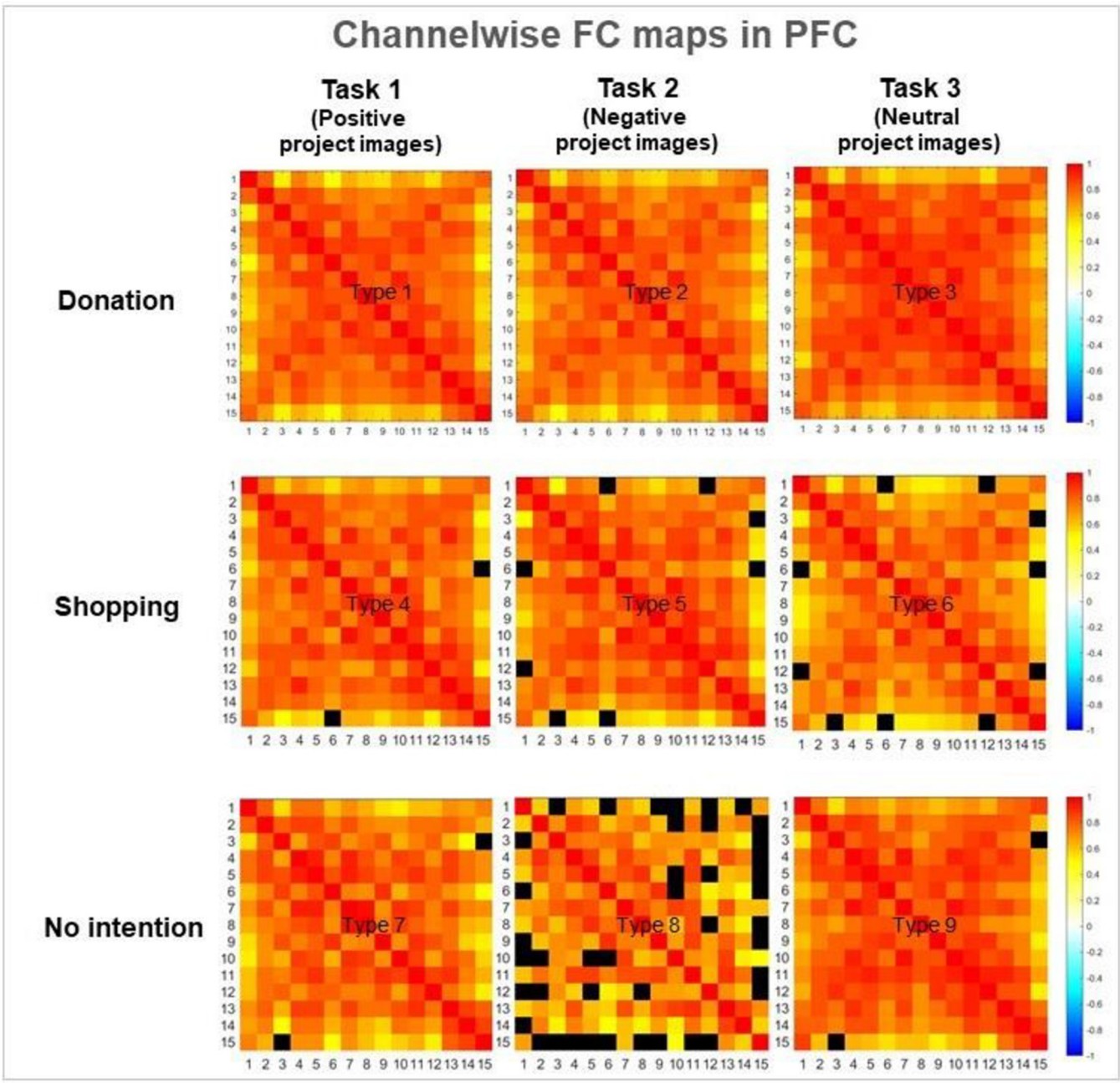

**Fig 5. Channelwise functional connectivity (FC) maps in the prefrontal cortex through grouped-averaged correlation matrices.** Each pixel in the correlation matrix maps has a Pearson correlation coefficient (r) of the responding channel pair. The horizontal axis indicates the willingness to donate; the vertical axis represents the type of experimental task. The color bar represents correlations from -1 (low correlation) to 1 (high correlation). A higher donation activity indicates higher FC correlations, implying that the donor's mental brain functions can be predicted and detected.

correlation map indicates that a higher FC correlation results in higher intentions to donate, whereas a lower FC correlation indicates lower intentions to donate or the inability to make decisions about donations. Among the nine experimental types, the correlation strength was high, in the following order: Types 3, 1, 2, 9, 4, 7, 5, 6, and 8. Strong interhemispheric linkages were observed in Type 3 but not 8. The color bars are quantifiable brain activation ranging from -1 (low activation) to 1 (high activation). Black boxes indicate no functional connection between sensor signals. These findings enable identifying areas where connections are stronger or weaker. When subjects desired to donate, we discovered no FC difference between positive and negative beneficiary images, suggesting that promoting images of a poor beneficiary to the donor is unnecessary to raise donations.

## Discussion

### Effect of emotional image on donation intention

Most studies have reported that negative images greatly influence helping behavior. Many researchers have found that contribution advertising with negative imagery raises more money than donation advertisements with positive images [5,6,18,19]. Specifically, the beneficiaries' negative facial expressions have been exploited to increase donations [6]. In contrast, some studies have yielded contradictory outcomes. Recent studies have argued that positive images can be more useful because these images influence positive consumer assessments and donations [10,20]. In addition, consumers are known to synchronize helping behavior by sympathizing with negative images [21], but constant exposure to these negative images can also increase negative attitudes toward the advertisements, decreasing helping behavior [22,23].

In line with this trend, donors who participated in this study made similar donation decisions regardless of viewing a positive or negative image, as presented in Fig 5. Additionally, we conducted a short interview with fifteen donors randomly among participants and found that the positive images in this experiment maximized their sympathy. Despite the positive expressions and bright impressions of the beneficiaries, the donors expressed regret about the poor conditions and situations. This phenomenon reveals that project images, whether positive or negative, that elicit individual empathy can improve the willingness to donate. Likewise, Bagozzi and Moor also found that sympathetic images elicited donation activity, regardless of the emotional images [4]. If sponsorship images inspire sympathy for donors, it is not necessary to use negative sponsorship images. However, this phenomenon is not observed in neutral images. Our research reveals that many participants reacted positively to the neutral image, but this did not lead to donations. It has been known that people tend to feel a neutral image positively or negatively depending on their current mood state [24]. In this study, it is impossible to control people's ever-changing emotions, so this phenomenon is considered beyond the scope of the study.

### Functional connectivity as biomarkers determining donation intentions

Brain FC, or the synchronization of geographically distant spontaneous neuronal activity, has been discovered in various brain systems [25–27]. In addition, FC is believed to reflect interactions between neuronal populations [28], and the correlation structure of spontaneous activity in FC maps can provide insight into the human brain's underlying functional architecture [29]. Furthermore, FC has been extensively used to define the neuronal disconnection of many neurological and psychological conditions [30].

We performed channelwise connectivity analyses within the PFC. The findings revealed changes in FC based on donation decisions, confirming the highest association between fNIRS channels in the donation behavior of participants, as represented in Fig 5. The nine

connectivity states (i.e., Types 1 to 9) were generated, demonstrating the FC differences depending on donation decisions. Those who decided to donate had strong FC correlations in all task types. The color bars represent quantified brain activation between -1 (low) and 1 (high). Black boxes indicate low channelwise connectivity. The scientific evidence suggests that these FC correlations can be developed as a biomarker capable of detecting and predicting donors and nondonors by focusing on the brain FC.

### Necessity for an innovative donation incentivepolicy to promote and incentivize donations

Compensatory psychology is heavily influenced by donation activities [31,32]. Recent research has discovered that, as donation culture spreads, strong activation of the medial PFC region in the frontal lobe occurs, which has a significant connection to the brain's reward circuit [19,33]. However, the evidence is not observed in these findings. We only demonstrated changes in brain activation based on donation decisions on three emotional project images. The medial parts including the overall PFC associated with the compensation circuit are not activated in our experiment because it may or may not affect the neurological compensation circuits depending on the intention to donate. In this experiment, we categorized participants as donors or shoppers according to their survey responses and discovered differences in brain activity between material reward through shopping and emotional reward through donation. This difference leads to the deactivation of the entire PFC, including the medial PFC, associated with neurological reward aspects, when a more valuable reward than any reward for donation is available.

These PFC activity differences are explained well in Fig 6. depending on the donation intention of the subjects who watched positive, negative, and neutral project images. Those who indicated a willingness to donate after watching the project images displayed PFC inactivation (Fig 6, left), whereas those who responded that they would shop (i.e., not donate) displayed PFC activation (Fig 6, right). This phenomenon is consistent with previous research results [34], indicating a decrease in frontal brain waves in people who make voluntary donations. Saffari *et al*. also confirmed that while shopping in a online shopping mall, people's brain activation increases [35]. Hence, people who do not intend to donate will require a new contribution policy because they will not profit much from the current donation incentives.

### Conclusion

We divided the hemodynamic signaling responses of 64 participants by their donation intentions into nine types and compared them with each other. The findings indicated that emotional project images did not affect the decision to help or not, regardless of whether the project images were positive or negative. It is consistent with the survey responses on intentions to donate after viewing project images in this experiment. However, it relies on the participants' presurvey donation intentions or their subsequent response to donation or shopping. The results imply that a negative project image does not always lead to donations. Moreover, more donation activity results in stronger FC correlations, implying that the donor's mental brain function could be predicted and recognized. This research can be useful in building future cause marketing strategies for charities and nonprofit organizations.

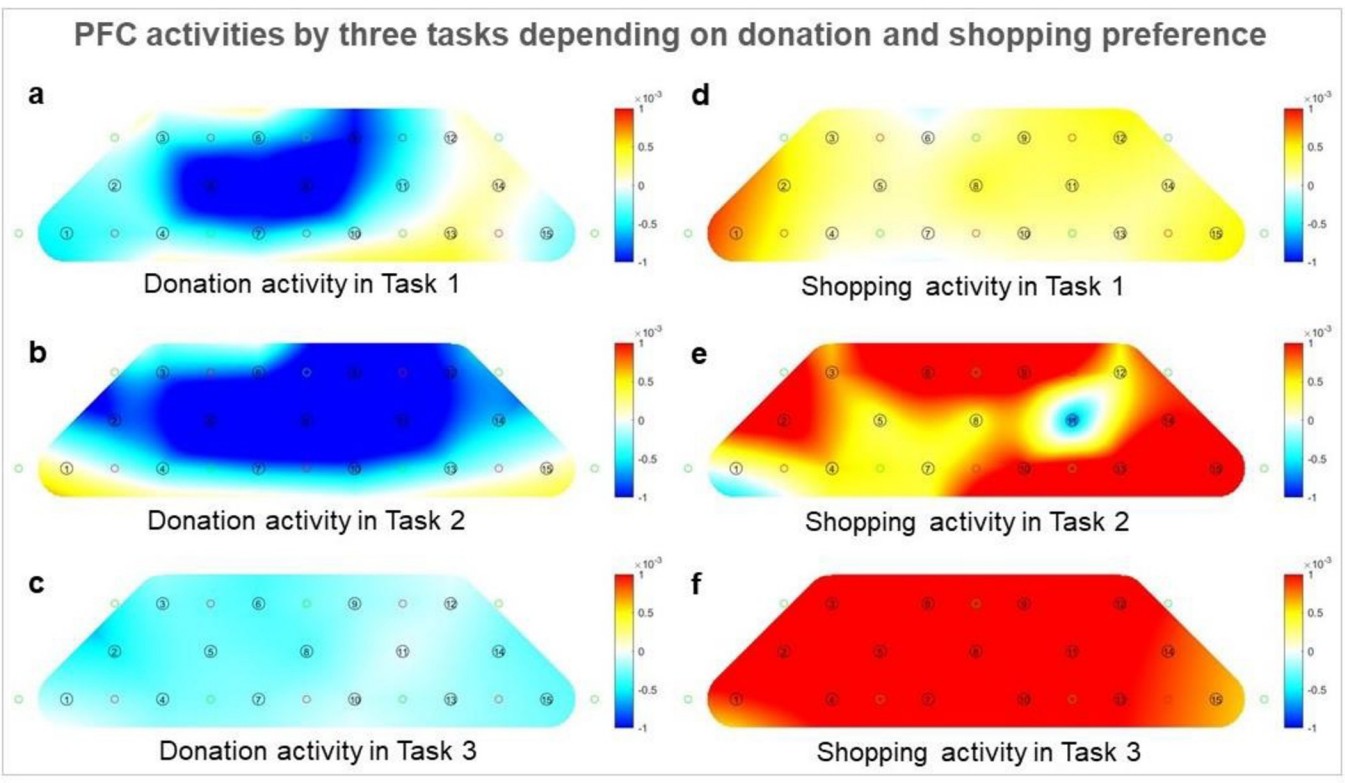

**Fig 6.** Prefrontal cortex (PFC) activities by three tasks depending on donation (left panel) vs. shopping (right panel) preference. (a) to (c) and (d) to (f) indicate Tasks 1 to 3, respectively, displaying the brain activity status of each randomized subject from (a) to (f). Channels 1–7 indicate the right PFC, and the rest indicate the left PFC. The color bar represents correlations from -1 (low correlation) to 1 (high correlation). Donation respondent brain activation is relatively lower than that of shopping respondents. This is slightly different from task to task, but the difference between PFC activities that respond to donation and shopping is evident after viewing the project images.

## Supporting information

**S1 File.**
(ZIP)

## Acknowledgments

We thank Essayreview (www.essayreview.co.kr) for performing the English language editing.

## Author Contributions

**Conceptualization:** SuJin Bak, Minsun Yeu, Jichai Jeong.

**Data curation:** SuJin Bak, Minsun Yeu, Jichai Jeong.

**Formal analysis:** Minsun Yeu, Dongwon Min, Jaehoon Lee.

**Funding acquisition:** SuJin Bak.

**Investigation:** Jaehoon Lee, Jichai Jeong.

**Methodology:** SuJin Bak, Minsun Yeu, Jichai Jeong.

**Project administration:** Dongwon Min, Jaehoon Lee, Jichai Jeong.

**Software:** SuJin Bak, Jaehoon Lee.

**Supervision:** SuJin Bak, Dongwon Min, Jaehoon Lee.

**Validation:** SuJin Bak, Jaehoon Lee.

**Visualization:** SuJin Bak, Jaehoon Lee.

**Writing – original draft:** SuJin Bak, Minsun Yeu, Dongwon Min, Jaehoon Lee, Jichai Jeong.

**Writing – review & editing:** SuJin Bak, Minsun Yeu, Dongwon Min, Jaehoon Lee, Jichai Jeong.

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
