## [Decision Letter · Decision Letter 0]

12 Feb 2024

PONE-D-23-35092Charitable crowdfunding donation-intention estimation depending on emotional project images using fNIRS-based functional connectivityPLOS ONE

Dear Dr. Bak,

Thank you for submitting your manuscript to PLOS ONE. After careful consideration, we feel that it has merit but does not fully meet PLOS ONE’s publication criteria as it currently stands. Therefore, we invite you to submit a revised version of the manuscript that addresses the points raised during the review process.

Dear authors,

firstly, I am sorry for the delay - but unfortunately it was not easy to find reviewers. I am happy that  now I can make a decision. I now have the report of a reviewer who is an expert in his/her field. Based on this report, I am happy to invite a major revision.

The report is very detailed and very helpful. In terms of having a clear path of improving the article, such that it can become publishable, I really expect that you clearly follow all these helpful comments made by the reviewer.

Based from my own reading of the article, I completely agree with the reviewer. One of the main issues is really that the experimental design is very hard to follow. I also asked myself about the timing of collecting the data on participants' intentions of donating or not. At the same time, the article is written in a manner that is very policy-related, which is good. However, this suggests that you may come up with policy implications (e.g., how to increase donation motivation by this image manipulations) and what are the potential underlying neuro-image channels? If for this reason, you also asked for the ex post willingness to donate (after treatment), then be clear about it and explain it.

In summary. three things are really important for this revision: (1) clearly outline your exact experimental design more comprehensively in the "Methods" section; (2) clearly and more openly discuss and admit the limitations of your design (e.g., also as outlined by the reviewer, when you report in your manipulation checks the rather low shares of people who agreed that the shown neutral images are really neutral); another issue is, if you really find a null effect - also discuss what this means. I am aware about the fact that in neuro studies the sample size is typically rather small. However, this still raises some questions about the validity of your findings. Is it really a null effect when the sample size is low? Be more cautious about interpreting that. (3) You report a couple of different findings and presentations in your figures. It would be good, if you could increase the focus on your main findings that you want to sell with this paper (e.g., the ones of Fig. 5). However, try to be cautious in terms of not overinterpreting the findings.

Good luck for the revision!

Best

Holger Rau

We look forward to receiving your revised manuscript.

Kind regards,

Holger A. Rau

Academic Editor

PLOS ONE

5. We note that Figure 1, 2, 4 and 6 in your submission contain copyrighted images. All PLOS content is published under the Creative Commons Attribution License (CC BY 4.0), which means that the manuscript, images, and Supporting Information files will be freely available online, and any third party is permitted to access, download, copy, distribute, and use these materials in any way, even commercially, with proper attribution. For more information, see our copyright guidelines: http://journals.plos.org/plosone/s/licenses-and-copyright.

a. You may seek permission from the original copyright holder of Figure 1, 2, 4 and 6 to publish the content specifically under the CC BY 4.0 license. 

Additional Editor Comments:

Dear authors,

firstly, I am sorry for the delay - but unfortunately it was not easy to find reviewers. I am happy that I can make a decision. I know have the report of a reviewer who is an expert in his/her field. Based on this report, I am happy to invite a major revision.

The report is very detailed and very helpful. In terms of having a clear path of improving the article, such that it can become publishable, I really expect that you clearly follow all these helpful comments made by the reviewer.

Based from my own reading of the article, I completely agree with the reviewer. One of the main issues is really that the experimental design is very hard to follow. I also asked myself about the timing of collecting the data on participants' intentions of donating or not. At the same time, the article is written in a manner that is very policy-related, which is good. However, this suggests that you may come up with policy implications (e.g., how to increase donation motivation by this image manipulations) and what are the potential underlying neuro-image channels? If for this reason, you also asked for the ex post willingness to donate (after treatment), then be clear about it and explain it.

In summary. three things are really important for this revision: (1) clearly outline your exact experimental design more comprehensively in the "Methods" section; (2) clearly and more openly discuss and admit the limitations of your design (e.g., also as outlined by the reviewer, when you report in your manipulation checks the rather low shares of people who agreed that the shown neutral images are really neutral); another issue is, if you really find a null effect - also discuss what this means. I am aware about the fact that in neuro studies the sample size is typically rather small. However, this still raises some questions about the validity of your findings. Is it really a null effect when the sample size is low? Be more cautious about interpreting that. (3) You report a couple of different findings and presentations in your figures. It would be good, if you could increase the focus on your main findings that you want to sell with this paper (e.g., the ones of Fig. 5). However, try to be cautious in terms of not interpreting the findings.

Good luck for the revision!

Best

Holger Rau

Reviewers' comments:

Reviewer's Responses to Questions

**Comments to the Author**

1. Is the manuscript technically sound, and do the data support the conclusions?

Reviewer #1: Partly

2. Has the statistical analysis been performed appropriately and rigorously? 

Reviewer #1: Yes

3. Have the authors made all data underlying the findings in their manuscript fully available?

Reviewer #1: Yes

4. Is the manuscript presented in an intelligible fashion and written in standard English?

Reviewer #1: Yes

5. Review Comments to the Author

Reviewer #1: The authors used fNIR spectroscopy to examine the controversy about whether images with positive or negative emotional valence are more effective in stimulating people to make donations on online crowdfunding platforms. They recruited 64 young adults (ages 20 to 33, half female) from Korea University with written informed consent, but do not describe what how subjects were recruited or what they were told about the experiment. They report that “per presurvey donation intentions, we divided the experimental participants into three groups: donation, no donation (shopping), and non-intention groups (Methods, page 4, lines 8-9).” They presented positive, negative, and neutral images to induce emotional states in participants before they made two types of donation decisions (to donate or to shop (Methods, page 4, line 82). All subjects were apparently exposed to all three types of images (positive, negative, and neutral), as shown by the total number of subjects who rated each type of image for Tasks 1-3 (see Table 1). However, this is not clearly described, raising concern about exposure to multiple types of images in all subjects on the results presented. Generally, the design is really not clearly described despite 6 Figures and 1 Table.

They say in conclusions (page 14-15) that “we divided the hemodynamic signaling responses of 64 participants by their donation intentions into nine types and compared them with each other/ They conclude that the valence of the images did not affect the decision to help or not, but it is not clear if this means it did not change presurvey donation intentions or the final response to donate or shop. They go on to say that more donation activity results in stronger functional connectivity in the prefrontal cortex, and it could provide a way that the donor’s mental brain function could be predicted and recognized. However, there are several issues that need to be clarified regarding the design of the study, the results, and the discussion.

It appears from Methods that the emotional images were generic, provided by the platform and are not tied specifically to unique charitable causes: “All project images used as donation images on Happybean were provided by the Korean internet platform Naver… (page 4, line 80). Then they presented positive, negative, and neutral emotional images to induce “incidental emotional states in participants before they made two types of donation decisions”, as shown in Figure 1 describing the overall experimental procedure (page 5 with 9 experimental types emerging from the possible combinations of 3 tasks (positive, negative, and neutral images), and (I think) presurvey intentions (donation, shopping, no intention). The survey results of the valence ratings using a 7 point Likert scale for the 3 sets of images (positive, negative, neutral are given in Table 1 on page 7. They summarize this by concluding that the “valence ratings of the project images in this study is properly designed”, but in fact the table makes clear that the neutral images were rated neutral only 36% (23 subjects) of the time and were rated positively by 40 subjects. Instead of pointing this out, they focus on the fact that the most frequent category in the seven-point scale is in conformity with their design, even though the overall direction of the valence is not so for the neutral images. They show in Figure 3 (results, page 8) that both positive and negative image sets increased the intensity of intention to donate or not (7 point Likert scale, not a dichotomous choice), whereas the images called neutral were mostly associated with not donating, even though they had been rated as positive by most subjects. What does this mean about the adequacy of the quantitative ratings of valence and intention to donate or shop? Something else is operating rather than just emotional valence of the images because most neutral images are rated positive by most subjects, but this is not discussed.

The authors found no effect of image valence on the hemodynamic activity of the prefrontal cortex, as illustrated in Figure 4. The Figure shows the intentions to donate (presumably by design this was the intention rated presurvey and pre-image exposure) on the horizontal axis and the vertical axis is the variation from reduced activity (-1, labeled as low activation) to increased activity (+1, labeled as high activation). Nearly the responses for all 9 types (intention x image valence) are for low activation (Figure 4, below 0, dark to light blue with some white for no change) except for a dash of yellow for type 8. In the discussion, the authors present more results illustrated in their Figure 6, showing that the decision to donate (presumably in response to exposure to valenced images) was associated with reduced prefrontal hemodynamic activity whereas the decision to shop was associated with increased prefrontal hemodynamic activity. More discussion is needed about the meaning of this difference. Are the positive and negative valenced images just weak and unexciting? Does donation generally induce a state of calm rest from engagement with a charitable cause? Is anticipation of shopping more rewarding than helping others because the subjects in this study were self-preoccupied and not altruistic? What can be said about the meaning of this study without knowing more about the personality and motivations of the subjects studied?

The authors have emphasized their findings about functional connectivity in relation to donation in their title and conclusions, and this is potentially their most interesting observation, as described in Figure 5. They say that “regardless of task type, the correlation map indicates that a higher FC correlation results in higher intentions to donate, whereas a lower FC correlation indicates lower intentions to donate or the inability to make decisions about donations.” However, inspection of the color bars show that nearly all 9 intention types have high functional connectivity, indicated red to yellow. There are channels with no detectable functional connectivity associated with shopping (why is this black instead of white as shown in the legend?). However, no statistical results are presented to indicate that the results are significant to justify the qualitative statements and conclusions.

In the discussion, Pages 13 and 14 (related to Figure 6), the authors state that donation has been associated with activation of the medial prefrontal cortex in other prior research, but that the activation of the medial prefrontal cortex “is not observed in our experiment.” (page 13, line 173). They go on to say that the observed deactivation of the entire PFC, including the medial PFC” (page 13, line 176). However, Figure 2 shows the placement of the fNIR spectroscopy channels, leads, and detectors, which appear to be mostly located in what is usually called the anterior cortex or frontal poles (BA 10). This includes anterior aspects of the medial PFC but it seems an overstatement to say that the entire PFC can be measured by such placement using fNIRS. Perhaps the authors could be more clear about what brain regions are actually measured in this experiment.

If the authors did record, subjects’ intentions prior to experimental manipulation, why didn’t they present any information about whether the baseline intentions were modified by their manipulation. That is, if someone set out to shop, was that changed by any of the experimental conditions? Or if they set out to donate, was that changed? If they had no preferred intention, did the intervention change that? Is the intention prior to experimental manipulation, a crude proxy for a measurement of their personality and whether they are self-centered, empathic, or charitable. This seems to be addressed to some extent by the results they report of short interviews of some donors after the experiment who reported that positive images had maximized their sympathy (page 12, line 149). Such a conclusion would require statistical evidence that positive images had changed intention to donate from its baseline following a planned intervention, but it is not clear that this is what the authors have reported. I cannot be sure because I still have questions about their design despite studying the paper carefully. This is an interesting study and I hope the authors will be able to answer my questions and obtain more data about who their subjects are in future work. As they say, the PFC has a key role in the regulation of emotional reactions, and its functional connectivity is strongly associated with individual differences in temperament and character (Zwir, I., Arnedo, J., Mesa, A. et al. Temperament & Character account for brain functional connectivity at rest. Mol Psychiatry (2023). https://doi.org/10.1038/s41380-023-02039-6).

6. PLOS authors have the option to publish the peer review history of their article (what does this mean?). If published, this will include your full peer review and any attached files.

Reviewer #1: No

---

## [Author Response · Author response to Decision Letter 0]

28 Mar 2024

We appreciate your contribution. According to your statement, we respond as follows. We respond by dividing your statement into nine questions temporarily. If we misunderstood your intention, please let us know again.

1) The authors used fNIR spectroscopy to examine the controversy about whether images with positive or negative emotional valence are more effective in stimulating people to make donations on online crowdfunding platforms. They recruited 64 young adults (ages 20 to 33, half female) from Korea University with written informed consent, but do not describe what how subjects were recruited or what they were told about the experiment. 

Response: We supplemented the manuscript as follows.

“In this study, 64 people (32 males and 32 females) from Korea University between 20 and 33 (mean and standard deviation of age: M = 25.67 sd = 2.69 years) participated in a same-day fNIRS study. All participants were recruited from the online community around KOPAS affiliated with Korea University in a month way. We have explained the whole process of experimenting, and all participants fully understood it.”

2) They report that “per presurvey donation intentions, we divided the experimental participants into three groups: donation, no donation (shopping), and non-intention groups (Methods, page 4, lines 8-9).” They presented positive, negative, and neutral images to induce emotional states in participants before they made two types of donation decisions (to donate or to shop (Methods, page 4, line 82). All subjects were apparently exposed to all three types of images (positive, negative, and neutral), as shown by the total number of subjects who rated each type of image for Tasks 1-3 (see Table 1). However, this is not clearly described, raising concern about exposure to multiple types of images in all subjects on the results presented. Generally, the design is really not clearly described despite 6 Figures and 1 Table.

Response: We supplemented the manuscript as follows.

“All subjects were reportedly exposed to all three types of images (positive, negative, and neutral), as specified in the within-subject design (i.e., repeated measures design) process. This design can minimize the variance caused by individual differences among participants.”

3) They say in conclusions (page 14-15) that “we divided the hemodynamic signaling responses of 64 participants by their donation intentions into nine types and compared them with each other/They conclude that the valence of the images did not affect the decision to help or not, but it is not clear if this means it did not change presurvey donation intentions or the final response to donate or shop. 

Response: We supplemented the manuscript as follows.

“However, it relies on the participants' presurvey donation intentions or their subsequent response to donation or shopping.”

4) They go on to say that more donation activity results in stronger functional connectivity in the prefrontal cortex, and it could provide a way that the donor’s mental brain function could be predicted and recognized. However, there are several issues that need to be clarified regarding the design of the study, the results, and the discussion. It appears from Methods that the emotional images were generic, provided by the platform and are not tied specifically to unique charitable causes: “All project images used as donation images on Happybean were provided by the Korean internet platform Naver… (page 4, line 80). Then they presented positive, negative, and neutral emotional images to induce “incidental emotional states in participants before they made two types of donation decisions”, as shown in Figure 1 describing the overall experimental procedure (page 5 with 9 experimental types emerging from the possible combinations of 3 tasks (positive, negative, and neutral images), and (I think) presurvey intentions (donation, shopping, no intention). The survey results of the valence ratings using a 7 point Likert scale for the 3 sets of images (positive, negative, neutral are given in Table 1 on page 7. They summarize this by concluding that the “valence ratings of the project images in this study is properly designed”, but in fact the table makes clear that the neutral images were rated neutral only 36% (23 subjects) of the time and were rated positively by 40 subjects. Instead of pointing this out, they focus on the fact that the most frequent category in the seven-point scale is in conformity with their design, even though the overall direction of the valence is not so for the neutral images. They show in Figure 3 (results, page 8) that both positive and negative image sets increased the intensity of intention to donate or not (7 point Likert scale, not a dichotomous choice), whereas the images called neutral were mostly associated with not donating, even though they had been rated as positive by most subjects. What does this mean about the adequacy of the quantitative ratings of valence and intention to donate or shop? Something else is operating rather than just emotional valence of the images because most neutral images are rated positive by most subjects, but this is not discussed.

Response: We agree with you. We didn't mean it, but we think this result is exaggerated, so it has been revised as below.

“On the other hand, Task 3 elicited predominanatly neutral responses, yet there was a notable majority expressing positive feedback. In this study, we are focused on the effect of positive and negative images on donations, and the valence quantitative ratings of positive and negative tasks are considered appropriate.”

Response: Additionally, we discussed these results as follows.

“However, this phenomenon is not observed in neutral images. Our research reveals that many participants reacted positively to the neutral image, but this did not lead to donations. It has been known that people tend to feel a neutral image positively or negatively depending on their current mood state [23]. In this study, it is impossible to control people's ever-changing emotions, so this phenomenon is considered beyond the scope of the study.”

5) The authors found no effect of image valence on the hemodynamic activity of the prefrontal cortex, as illustrated in Figure 4. The Figure shows the intentions to donate (presumably by design this was the intention rated presurvey and pre-image exposure) on the horizontal axis and the vertical axis is the variation from reduced activity (-1, labeled as low activation) to increased activity (+1, labeled as high activation). Nearly the responses for all 9 types (intention x image valence) are for low activation (Figure 4, below 0, dark to light blue with some white for no change) except for a dash of yellow for type 8. In the discussion, the authors present more results illustrated in their Figure 6, showing that the decision to donate (presumably in response to exposure to valenced images) was associated with reduced prefrontal hemodynamic activity whereas the decision to shop was associated with increased prefrontal hemodynamic activity. More discussion is needed about the meaning of this difference. Are the positive and negative valenced images just weak and unexciting? Does donation generally induce a state of calm rest from engagement with a charitable cause? Is anticipation of shopping more rewarding than helping others because the subjects in this study were self-preoccupied and not altruistic? What can be said about the meaning of this study without knowing more about the personality and motivations of the subjects studied?

Response: The purpose of this study is not to investigate the brain condition according to the subject's personality and motivation. The purpose of this study is to prove that there is little correlation between donation intention and brain activity between negative and positive project images. Also, the main focus of this section is to show the difference in brain activation between donation and shopping. Thus, we discussed the prior studies, and added it as follows;

“This phenomenon is consistent with previous research results [32], indicating a decrease in frontal brain waves in people who make voluntary donations. Saffari et al. also confirmed that while shopping in a online shopping mall, people's brain activation increases [33].”

6) The authors have emphasized their findings about functional connectivity in relation to donation in their title and conclusions, and this is potentially their most interesting observation, as described in Figure 5. They say that “regardless of task type, the correlation map indicates that a higher FC correlation results in higher intentions to donate, whereas a lower FC correlation indicates lower intentions to donate or the inability to make decisions about donations.” However, inspection of the color bars show that nearly all 9 intention types have high functional connectivity, indicated red to yellow. There are channels with no detectable functional connectivity associated with shopping (why is this black instead of white as shown in the legend?). 

Response: Our response is as follows.

White here is not a channel without functional connectivity, it just means the mid-range between high and low blood flow.

7) However, no statistical results are presented to indicate that the results are significant to justify the qualitative statements and conclusions.

Response: The functional connectivity (FC) method is already a principle that works based on statistics. Specifically, we explain this as follows

“FC works based on statistics, which maps the correlation between responses shown in the prefrontal cortex of a subject brain.”

8) In the discussion, Pages 13 and 14 (related to Figure 6), the authors state that donation has been associated with activation of the medial prefrontal cortex in other prior research, but that the activation of the medial prefrontal cortex “is not observed in our experiment.” (page 13, line 173). They go on to say that the observed deactivation of the entire PFC, including the medial PFC” (page 13, line 176). However, Figure 2 shows the placement of the fNIR spectroscopy channels, leads, and detectors, which appear to be mostly located in what is usually called the anterior cortex or frontal poles (BA 10). This includes anterior aspects of the medial PFC but it seems an overstatement to say that the entire PFC can be measured by such placement using fNIRS. Perhaps the authors could be more clear about what brain regions are actually measured in this experiment.

Response: We supplemented the manuscript as follows

“The medial parts including the overall PFC associated with the compensation circuit are not activated in our experiment because it may or may not affect the neurological compensation circuits depending on the intention to donate.”

9) If the authors did record, subjects’ intentions prior to experimental manipulation, why didn’t they present any information about whether the baseline intentions were modified by their manipulation. That is, if someone set out to shop, was that changed by any of the experimental conditions? Or if they set out to donate, was that changed? If they had no preferred intention, did the intervention change that? Is the intention prior to experimental manipulation, a crude proxy for a measurement of their personality and whether they are self-centered, empathic, or charitable. This seems to be addressed to some extent by the results they report of short interviews of some donors after the experiment who reported that positive images had maximized their sympathy (page 12, line 149). Such a conclusion would require statistical evidence that positive images had changed intention to donate from its baseline following a planned intervention, but it is not clear that this is what the authors have reported. I cannot be sure because I still have questions about their design despite studying the paper carefully. This is an interesting study and I hope the authors will be able to answer my questions and obtain more data about who their subjects are in future work. As they say, the PFC has a key role in the regulation of emotional reactions, and its functional connectivity is strongly associated with individual differences in temperament and character (Zwir, I., Arnedo, J., Mesa, A. et al. Temperament & Character account for brain functional connectivity at rest. Mol Psychiatry (2023). https://doi.org/10.1038/s41380-023-02039-6).

Response: The presurvey conducted in this experiment is a series of studies that lead to brain measurement experiments as soon as participants respond. As a result, it is carried out with the assumption that the subject's intention has not changed. Before experimental manipulation, it is a proxy for brain signal measurement by participants' propensity. This is supported by the results of short interviews with some donors after the experiments, showing that positive images maximize empathy. These short interviews are one of our detailed findings. However, Interviews were conducted with 15 donors at random among participants, and statistical results are unfortunately non-existent. Instead, we modified these as follows.

“We conducted a short interview with fifteen donors randomly among participants and found that the positive images in this experiment maximized their sympathy”

* Also, we cited the mentioned paper as a reference [12] to support for our research results.

---

## [Decision Letter · Decision Letter 1]

22 Apr 2024

Charitable crowdfunding donation-intention estimation depending on emotional project images using fNIRS-based functional connectivity

PONE-D-23-35092R1

Dear Dr. Bak,

We’re pleased to inform you that your manuscript has been judged scientifically suitable for publication and will be formally accepted for publication once it meets all outstanding technical requirements.

Kind regards,

Holger A. Rau

Academic Editor

PLOS ONE

Additional Editor Comments (optional):

I follow the recommendation of the reviewer to accept the article.** However, please address a last clarification mentioned by the reviewer: "The authors have clarified the questions raised in my initial review adequately. However, on page 4, line 86 there refer to "in a month way". It is unclear what this means: does it mean they advertized for volunteers for one month? This can be clarified in preparing the article in its final form."**

You should do this before you submit your final version of the manuscript to the journal.

Congrats and all the best!

Holger

Reviewers' comments:

Reviewer's Responses to Questions

**Comments to the Author**

1. If the authors have adequately addressed your comments raised in a previous round of review and you feel that this manuscript is now acceptable for publication, you may indicate that here to bypass the “Comments to the Author” section, enter your conflict of interest statement in the “Confidential to Editor” section, and submit your "Accept" recommendation.

Reviewer #1: All comments have been addressed

2. Is the manuscript technically sound, and do the data support the conclusions?

Reviewer #1: (No Response)

3. Has the statistical analysis been performed appropriately and rigorously? 

Reviewer #1: Yes

4. Have the authors made all data underlying the findings in their manuscript fully available?

Reviewer #1: Yes

5. Is the manuscript presented in an intelligible fashion and written in standard English?

Reviewer #1: Yes

6. Review Comments to the Author

Reviewer #1: The authors have clarified the questions raised in my initial review adequately.

Howevr, on page 4, line 86 there refer to "in a month way". It is unclear what this means: does it mean they advertized for volunteers for one month? This can be clarified in preparing the article in its final form.

7. PLOS authors have the option to publish the peer review history of their article (what does this mean?). If published, this will include your full peer review and any attached files.

Reviewer #1: No

---

## [Editor Report · Acceptance letter]

26 Apr 2024

PONE-D-23-35092R1 

PLOS ONE

Dear Dr. Bak, 

I'm pleased to inform you that your manuscript has been deemed suitable for publication in PLOS ONE. Congratulations! Your manuscript is now being handed over to our production team.

Kind regards, 

on behalf of

Prof. Dr. Holger A. Rau 

Academic Editor

PLOS ONE